# The Equity and Effectiveness of Achieving Canada’s Voluntary Sodium Reduction Guidance Targets: A Modelling Study Using the 2015 Canadian Community Health Survey—Nutrition

**DOI:** 10.3390/nu13030779

**Published:** 2021-02-27

**Authors:** Brendan T. Smith, Salma Hack, Mahsa Jessri, JoAnne Arcand, Lindsay McLaren, Mary R. L’Abbé, Laura N. Anderson, Erin Hobin, David Hammond, Heather Manson, Laura C. Rosella, Douglas G. Manuel

**Affiliations:** 1Health Promotion, Chronic Disease and Injury Prevention, Public Health Ontario, Toronto, ON M5G 1V2, Canada; salma.hack@oahpp.ca (S.H.); erin.hobin@oahpp.ca (E.H.); heather.manson@utoronto.ca (H.M.); 2Dalla Lana School of Public Health, University of Toronto, Toronto, ON M5T 3M7, Canada; laura.rosella@utoronto.ca; 3Food, Nutrition and Health Program, University of British Columbia, Vancouver, BC V6T 1Z4, Canada; mahsa.jessri@ubc.ca; 4Faculty of Health Sciences, University of Ontario Institute of Technology, Oshawa, ON L1H 7K4, Canada; JoAnne.Arcand@ontariotechu.ca; 5Department of Community Health Sciences, University of Calgary, Calgary, AB T2N 4Z6, Canada; lmclaren@ucalgary.ca; 6Department of Nutritional Sciences, University of Toronto, Toronto, ON M5S 1A8, Canada; Mary.Labbe@utoronto.ca; 7Department of Health Research Methods, Evidence, and Impact, McMaster University, Hamilton, ON L8S 4L8, Canada; ln.anderson@mcmaster.ca; 8Child Health Evaluative Sciences, Sickkids Research Institute, Toronto, ON M5G 0A4, Canada; 9School of Public Health and Health Systems, University of Waterloo, Waterloo, ON N2L 3G1, Canada; david.hammond@uwaterloo.ca; 10ICES, Toronto, ON K1Y 4E9, Canada; dmanuel@ohri.ca; 11Ottawa Hospital Research Institute, Clinical Epidemiology, Ottawa, ON K1H 8L6, Canada; 12Health Analysis Division, Statistics Canada, Ottawa, ON K1A 0T6, Canada; 13Department of Family Medicine, and School of Epidemiology and Public Health, University of Ottawa, Ottawa, ON K1H 8L6, Canada; 14Bruyère Research Institute, Ottawa, ON K1R 6M1, Canada

**Keywords:** sodium reformulation, sodium intake, 2015 Canadian Community Health Survey—Nutrition, social inequities, socioeconomic position, sodium reduction guidance targets

## Abstract

**Background**: High sodium intake is a leading modifiable risk factor for cardiovascular diseases. This study estimated full compliance to Canada’s voluntary sodium reduction guidance (SRG) targets on social inequities and population sodium intake. **Methods**: We conducted a modeling study using *n* = 19,645, 24 h dietary recalls (Canadians ≥ 2 years) from the 2015 Canadian Community Health Survey—Nutrition (2015 CCHS-N). Multivariable linear regressions were used to estimate mean sodium intake in measured (in the 2015 CCHS-N) and modelled (achieving SRG targets) scenarios across education, income and food security. The percentage of Canadians with sodium intakes above chronic disease risk reduction (CDRR) thresholds was estimated using the US National Cancer Institute (NCI) method. **Results**: In children aged 2–8, achieving SRG targets reduced mean sodium intake differences between food secure and insecure households from 271 mg/day (95%CI: 75,468) to 83 mg/day (95%CI: −45,212); a finding consistent across education and income. Mean sodium intake inequities between low and high education households were eliminated for females aged 9–18 (96 mg/day, 95%CI: −149,341) and adults aged 19 and older (males: 148 mg/day, 95%CI: −30,327; female: −45 mg/day, 95%CI: −141,51). Despite these declines (after achieving the SRG targets) the majority of Canadians’ are above the CDRR thresholds. **Conclusion**: Achieving SRG targets would eliminate social inequities in sodium intake and reduce population sodium intake overall; however, additional interventions are required to reach recommended sodium levels.

## 1. Introduction

Excess sodium intake is a leading modifiable risk factor for hypertension, cardiovascular diseases and diet-related mortality [1,2]. Sodium intakes in most high- income countries have exceeded the World Health Organization (WHO) guidelines of less than 2000 mg/day, leading to global initiatives targeting a 30% reduction in population-level sodium intake by 2025 [3]. Voluntary sodium reformulation targets directed at the food industry have been identified as a WHO “best buy” intervention due to its success in reducing population sodium intake in high-income countries [4,5]. As many as 38 countries have implemented sodium reformulation targets for foods as part of sodium reduction initiatives [6]. However, evidence from systematic reviews highlights poor monitoring and evaluation of sodium reformulation strategies, and insufficient information to assess differential impact of dietary sodium reduction interventions across socioeconomic factors as barrier to fully understanding their potential to reduce population sodium intake [5,7].

In Canada, approximately 58% of the population consumed above 2300 mg sodium/day in 2015 [8], the recommended maximum threshold for chronic disease risk reduction (CDRR) in a healthy population [9]. In 2012, the Canadian Government published voluntary sodium reduction guidance (SRG) targets designed to encourage a “gradual” sodium reduction of 25–30% in processed foods by 2016 [10]. At this time, 77% of Canadians overall sodium intake came from processed foods [8]. Despite setting SRG targets in consultation with the food industry, their compliance remained low in Canada [11]. In 2017, only 14% of processed foods targeted by the intervention achieved the 25–30% sodium reduction goal (48% did not make meaningful progress), and in some cases sodium increased [11].

Sodium interventions, such as SRG targets, have the potential to equitably reduce population sodium intake, especially if uptake is high [5,12,13]. These interventions aim to reduce sodium intake by targeting the environments in which behaviours occur rather than individuals’ behavioral change, in this case reformulating food composition compared to changing individuals’ diets [5,13]. Not relying on personal resources to reduce sodium intake (i.e., low agency interventions) removes an important barrier for designing equitable population-level interventions [14]. However, poor compliance to voluntary SRG targets may exacerbate existing or even generate sodium intake inequities by age, sex and socioeconomic position (SEP), in particular if reformulation is not consistently achieved across targeted food categories and brands consumed differentially across population sub-groups. Achieving full compliance, equivalent to setting mandatory reformulation targets, can limit this unintended effect by removing industry disincentives to comply with regulations because of the potential for a loss in competitive advantage. For example, a modelling study has shown narrowed social inequities in the UK with mandatory compared to voluntary sodium reformulation [15]. Furthermore, implementing mandatory sodium reformulation has been effective in reducing population sodium intake as seen in South Africa and Argentina [15,16,17,18,19].

Low socioeconomic position is associated with increased sodium intake in a systematic review and meta-analysis of high-income countries [20]. However, prior to SRG targets being introduced in Canada, there was no evidence of social inequities in sodium intake in either of the last two national nutrition surveys in 1970–72 and 2004, where high sodium intake was observed across all SEP groups [21]. Currently, no study has assessed the impact of poor compliance with SRG targets on social inequities in sodium intake. Data from the 2015 Canadian Community Health Survey—Nutrition (2015 CCHS-N), the first national dietary intake survey since 2004, provide an important opportunity to examine whether social inequities in sodium intake exist and the potential equitable impact of achieving SRG targets on sodium intakes in Canada. 

Therefore, the study objectives were to model the impact of achieving full compliance with Canada’s SRG targets on social inequities in sodium intake and the percentage of Canadians above the sodium intake CDRR threshold in a population-representative sample. This is critical to understanding the potential of national SRG targets to eliminate or avoid the generation of social inequities and reduce population sodium intake [13,20].

## 2. Materials and Methods 

### 2.1. Study Design and Population

A modelling study was conducted using data from the 2015 Canadian Community Health Survey—Nutrition (2015 CCHS-N). The 2015 CCHS-N is a nationally representative cross-sectional survey conducted by Statistics Canada to assess dietary intakes of Canadians using interviewer-administered 24 h dietary recalls. This survey used a multi-stage, cluster sampling approach to secure a sample of 20,487 Canadians aged 1 year and older living in private dwellings across the 10 provinces [22]. A subset of respondents (35%) were asked to complete a second 24 h dietary recall on a different day of the week [22]. The response rate was 62% for the first 24 h recall and 69% for the second day recall [22]. Interviews were conducted by proxy for children aged 1 to 5, jointly with the parent or guardian for children aged 6 to 11, and respondents aged 12 and older completed the interview independently [22].

In this study, sodium intake was assessed among Canadians aged 2 and older who participated in at least one 24 h dietary recall. Respondents were excluded if they were pregnant, breastfeeding, or did not report energy intake (*n* = 691), or had missing data on education (*n* = 41) or food security (*n* = 110). The final analytic sample included 19,645 Canadians. 

### 2.2. Sodium Intake

An adapted version of the Automated Multi-Pass Method from the US Department of Agriculture was used to collect 24 h dietary recalls, described elsewhere [22]. Trailing questions were also asked about the type and frequency of salt added by the consumer. However, the amount of added salt was not quantified and therefore could not be included in our analyses.

The nutritional composition of reported foods in the 2015 CCHS-N was estimated using the Canadian Nutrient File (CNF) 2015, a standard nutrient composition database continuously updated to reflect average nutrient values of foods regularly consumed by Canadians [22]. Nutrient values for foods not found in the CNF but reported in the 24 h dietary recall were estimated based on the recipe and survey foods file. 

### 2.3. Socioeconomic Position

Highest level of household education was categorised using the following four groups: “less than high school”, “high school”, “certificate from a trade, college, or non-bachelor certificate” and “bachelor degree or higher”. Adjusted household income quintiles were derived based on the ratio of respondents’ total household income reported in the last 12 months to Canada’s low-income cut-off corresponding to their household and community size [22]. Household food security was assessed using eight questions for children and 10 questions for adults (≥19 years), and responses were classified as either food secure (i.e., answered “yes” to 0–1 questions about difficulty with income-related food access) or food insecure (moderate or severe, i.e., answered “yes” to two or more questions about compromised quality/quantity or reduced food intake due to disrupted eating patterns) [22,23]. 

### 2.4. Covariates

To maximize the sample in each SEP group, Dietary Reference Intake age categories were combined as follows: children 2–8 years, 9–18 years and adults ≥19 years. A continuous age variable was included in models to control for residual confounding related to the broad age groups. Sex was defined as male or female. Total daily energy intake (continuous) was defined as energy intake from all food sources (kcal/day) [22]. 

Additionally, we adjusted for dietary misreporting to account for systematic error, a common phenomenon in self-reported dietary assessments where respondents under-report socially undesirable foods, often associated with increased sodium intake [24]. In children under age 12, dietary misreporting was categorised based on the ratio of total reported energy intake (EI) to total estimated energy requirements (EER) using established categories: under-reporters (EI:EER < 74%), plausible reporters (74% ≤ EI:EER ≤ 135%) or over-reporters (EI:EER > 135%) [24,25]. For individuals aged 12 and older, dietary misreporting was categorised using established cut-points: under-reporters (EI:EER < 70%), plausible reporters (70% ≤ EI:EER ≤ 142%) and over-reporters (EI:EER > 142%) [26]. EER was estimated using the US Institute of Medicine’s factorial equations, incorporating age, sex, self-reported physical activity, height and weight [22]. Respondents were grouped as “unclassified” if they could not be categorised due to missing information or reported as underweight. Refer to Appendix A for more details.

### 2.5. Modelling the Impact of Health Canada’s Sodium Reduction Guidance on Sodium Intake

Reported foods were regrouped into 15 categories and 94 sub-categories outlined in the SRG targets [11]. Coding was validated by three research team members and disputes (<1%) were resolved by consulting a registered dietitian. Foods that did not fit into the SRG sub-categories or foods without targets, such as ready-to-eat fresh fruits and vegetables, restaurant foods or homemade foods, were grouped into a separate “non-SRG” category. In the full compliance to SRG target scenario, sodium intake was estimated by setting sodium values in reported foods to those proposed in the final phase of the SRG targets (i.e., 25–30% reduction [11]). For each respondent, predicted total daily sodium intake was estimated by summing the sodium consumed in both SRG foods, estimated by multiplying the reported weight of foods consumed by the respective sodium target, and non-SRG foods.

### 2.6. Statistical Analyses

Adjusted mean sodium intake and mean differences and 95% confidence intervals (CI) were estimated using multivariable linear regression (with the LS means statement), controlling for continuous age, sex (in children aged 2–8), total daily energy intake and dietary misreporting. Adjusted means represent sodium intake on any given day and were estimated both as measured (no intervention) and as a full compliance to SRG targets scenario. Separate models were run for sex-pooled (aged 2–8) and sex-specific (aged 9–18 and aged 19 and older) analyses for each SEP indicator. 

The US National Cancer Institute (NCI) method was used to estimate the distribution of usual sodium intake, and the percentage of respondents who had sodium intakes above the CDRR threshold, both in the measured and the modelled full compliance to SRG targets scenario [27]. The distribution of usual sodium intake was modelled using 100 Monte Carlo simulations (using the DISTRIB macro), including a subset of respondents’ second day recalls (*n* = 7381) to account for between- and within-person variations in dietary intake, with additional adjustment for age, dietary misreporting, order of recall (i.e., recall 1 or 2) and weekend or weekday dietary record [22,27]. Excess sodium intake was based on the highest CDRR threshold in a corresponding age group. The percentage of respondents with sodium intake above the CDRR threshold was estimated by sex and SEP groups for respondents aged 2–8 (1500 mg/day), aged 9–18 (2300 mg/day) and aged 19 years and older (2300 mg/day) [8,9].

Following Statistics Canada procedures [22], all analyses were weighted and 95% CIs were bootstrapped using 500 balanced repeated replications in SAS 9.4. This study was approved by the Ethics Review Board at Public Health Ontario. 

## 3. Results

The final analytic sample included 19,645 Canadians. Unadjusted mean sodium intakes across respondent characteristics are presented in Table 1. Mean sodium intake was higher in males than in females in their corresponding age groups and consistently above 2300 mg/day in all male age groups. Mean sodium intake in children (aged 2–18) was highest among households with less than high school education and food insecurity. However, in children, the highest income quintile had higher sodium intake compared to all lower quintiles. In adults, the highest income quintile and food secure groups had higher sodium intake than lower income quintiles and food insecure groups.

### SRG Targets and Social Inequities in Sodium Intake

For children aged 2–8, adjusted mean sodium intake across SEP groups is shown in Table 2. Measured mean sodium intake was consistently higher in children from low SEP compared to high SEP households. Mean sodium intake was 358 mg/day (95%CI: −201,916) higher in children with “less than high school” compared to “bachelor degree or higher” household education, and 170 mg/day (95%CI: −1342) higher in children from households in the lowest (quintile 1) compared to highest income quintile (quintile 5). Considerable uncertainty exists across these two estimates resulting from smaller samples of respondents aged 2–8; however, large absolute differences in children’s mean sodium intake between SEP groups were observed. Mean sodium intake was 271 mg/day (95%CI: 75,468) higher in children from food insecure compared to food secure households. Social inequities in sodium intake, the absolute differences observed across SEP groups, were eliminated in the modelled full compliance intervention scenario. 

For respondents aged 9–18, the sex-specific adjusted mean sodium intakes across SEP groups are shown in Table 3. As measured, social inequities in mean sodium intake were inconsistent. Sodium intake was 148 mg/day (95%CI: −161,456) and 246 mg/day (95%CI: −1493) higher for males and females, respectively, with “less than high school” compared to “bachelor degree or higher” household education. Conversely, sodium intake was lower in respondents from households in the lowest (quintile 1) compared to the highest income quintile (quintile 5) for males (224 mg/day, 95%CI: −442, −5), with no differences observed for females. Mean sodium intake was similar across food security groups for males and females. Social inequities in mean sodium intake were eliminated in the modelled full compliance intervention scenario.

For adults aged 19 and older, sex-specific adjusted mean sodium intakes across SEP are presented in Table 3. As measured, adults in households with “less than high school” education had higher mean sodium intake compared to those with “bachelor degree or higher” for males (399 mg/day, 95%CI: 172,626) and females (122 mg/day, 95%CI: 6239). No differences in measured mean sodium intake were observed across income or food insecurity groups in males or females. In the modelled full compliance intervention scenario, observed educational inequities in adjusted mean sodium intake between adults with “less than high school” compared to “bachelor degree or higher” household education were reduced by half for males (148 mg/day, 95%CI: −30,327) and eliminated for females (−45 mg/day, 95%CI: −14,151). Further, adjusted mean sodium intake was reduced to a greater extent in adults from food insecure compared to food secure households, generating an inverse association between food insecurity and sodium intake for both males and females. 

#### SRG Targets and Usual Sodium Intake Distributions

Usual sodium intake distributions for each age and sex group are presented in Figure 1. Achieving full compliance with SRG targets was estimated to both reduce sodium intake across the distributions (shifting the curve towards lower usual sodium intake values) and also among the highest sodium consumers (increasing the peak and narrowing the distribution). 

The percentage of Canadians above the CDRR threshold across age/sex and SEP groups is presented in Appendix A. In the modelled full compliance scenario, the percentage of Canadians above the CDRR threshold remained high overall, and higher in males than in females, despite the observed decline compared to measured sodium intake. For example, 79% (95%CI: 74, 84) of children aged 2–8, 76% (95%CI: 71, 82) of males and 48% (95%CI: 42, 54) of females aged 9–18, and 71% (95%CI: 67, 74) of adult males and 34% (95%CI: 3137) of adult females had sodium intakes above their respective CDRR threshold. Similar patterns were observed across SEP groups, where the percentage of Canadians with sodium intake above the CDRR thresholds remained high in the modelled full compliance intervention scenario but with a consistent decline observed across all SEP indicators for each age group examined. 

## 4. Discussion

Using data from a nationally representative sample of the Canadian population, this study modelled the equity and effectiveness of fully achieving national voluntary SRG targets for processed foods. Overall, fully achieving the SRG targets was demonstrated to eliminate measured social inequities in sodium intake and reduce population sodium intake. This is a critical finding, as for the first time in Canada, the measured sodium intakes were observed to be higher among individuals in lower compared to higher SEP across all age groups [21,28]. Of concern, the majority of Canadians (specifically males) would continue to have sodium intakes above the advisable CDRR thresholds if SRG targets were fully achieved.

The present study demonstrates the potential of full compliance to national SRG targets for eliminating existing social inequities in measured mean sodium intakes in Canada across all age groups. Achieving the full SRG targets in Canada, without additional behavioural change interventions, was sufficient to eliminate social inequities in sodium intake. A previous modelling study in England estimated mandatory reformulation (equivalent to modelled full compliance to voluntary SRG targets in Canada) would achieve larger reductions in low compared to high SEP groups, narrowing sodium intake inequities [15]. Our study’s results further confirm these findings highlighting the role of poor compliance in eroding equitable outcomes of sodium reformulation interventions. Together, these nationally representative studies underscore the importance of low agency interventions for equitably reducing population sodium intake and existing social inequities in intake [15]. In addition, these studies emphasize the effectiveness of full compliance, or mandatory regulations, versus voluntary approaches for generating more equitable sodium intake reductions by eliminating food manufacturer disincentives to reach targets, and limiting industry choice (or agency) regarding where and when to comply with targets [5].

Social inequities in sodium intake observed in our study add high-quality nationally representative Canadian evidence to a mixed literature. A meta-analysis estimated low compared to high SEP groups consumed an additional 503 mg/day (95%CI: 461,545) of sodium in studies that used urine-based methods to assess sodium intake, although this association was not consistent across all studies that used urine-based or dietary recalls to assess sodium intake [20]. Previous Canadian studies using the 2004 CCHS (the previous national nutrition survey) found no association between household food security and sodium intake in Canadians aged 2–18 [28] or between education, income or food security and sodium intake in adults [21,28]. The present study observed social inequities in measured sodium intake across education, income and food security in children aged 2–8, and education in males and females aged 9–18. These results suggest that childhood is a sensitive period where increased sodium intake may be related to both available household knowledge (education) and material resources (education, income and food security) [29]. Among adults, this study reveals that low household education, on average, consumes more sodium than those with high household education, but also confirmed previous findings of no association between income or food security and sodium intake. Although context dependent, proposed pathways between low SEP and higher sodium intake include poor diet quality resulting from the ubiquitous marketing of unhealthy foods (high in sodium) to children, or the financial cost associated with healthy food choices [14,30,31]. Differences in findings between Canadian nutrition surveys may be due to the introduction of SRG targets, changes to the food supply and dietary patterns over time or methodological differences limiting survey comparability.

Our study showed achieving SRG targets alone would be insufficient to bring the majority of Canadians’ usual sodium intake below the CDRR threshold. As hypothesized, our model confirmed previous evidence from the US demonstrating that full compliance to sodium reformulation targets for processed foods would shift (i.e., narrow and reduce) usual sodium intake distribution curves [32]. However, our findings suggest that additional sodium interventions are necessary to reduce population-level sodium intake to help curb the rise in chronic diseases and improve dietary intakes of Canadians. Actions to increase compliance, for example, through government leadership or implementation of mandatory sodium targets for key foods are warranted. A systematic review of global dietary salt policies concluded that implementing a comprehensive, multifaceted sodium reduction strategy would be the most successful approach to reduce population sodium intake [18]. Following this advice, it is proposed that updated sodium reduction strategies combine structural interventions (such as mandatory and more stringent reformulation of foods identified as top sodium contributors such as bakery products, mixed dishes, processed meats [8], etc., and sodium targets for other food industry sectors such as restaurant and food services), jointly with downstream interventions (such as educational awareness campaigns and front-of-package labels) [5,18]. However, implementing such strategies will require a delicate balance of achieving not only a reduction in population-level sodium intake but also an equitable reduction in sodium for all Canadians, as observed in this study.

This study is not without its limitations. First, our approach does not account for potential changes in consumer behaviour (e.g., the type or amounts of products consumed) that may occur as a result of sodium reformulation. These behaviours could moderate or enhance the estimated changes to sodium intake from reformulation alone, which are modeled in the current study. Second, sodium intakes may be underestimated as they do not account for table salt added by the consumer, estimated to be an additional 10% in Canada [8,22]. Third, the use of 24 h dietary recalls, compared to urinary sodium intake measures, remains a contentious topic as there is no exact method to capture all sources of dietary sodium [2,9]. Sodium is likely underestimated in this study, however, in the absence of the gold standard (multiple urinary collections recovering 90–92% of dietary sodium), lack of feasibility and cost associated with data collection for nationally representative studies, the 24 h dietary recall data in the 2015 CCHS-N provide the best available measure of population-level sodium intake in Canada [2,9,33]. Fourth, reported foods could not be linked to branded information and therefore sodium levels reported reflect averages rather than true sodium values from a specific brand. As brand choice is socially patterned and often influenced by price with sodium reduced products often sold as premium brands, social inequities in sodium intake in this study are likely conservative [28,29].

Strengths of this study include providing up-to-date Canadian representative estimates of population and social inequities in sodium intake. Furthermore, an equity lens is added to the analysis to demonstrate the potential of SRG targets to reduce sodium intake in the population, and also across age, sex and SEP. The NCI method was applied to examine usual sodium distributions and estimate the percentage of Canadians above CDRR overall and across population subgroups.

## 5. Conclusions

This study highlights the cost of inaction in Canada towards meeting the voluntary SRG sodium targets in processed foods, with marked social inequities in sodium intake and persistent high population-level sodium intake. Achieving SRG targets has the potential to eliminate existing sodium intake inequities and reduce population sodium intake; however, the majority of Canadian’s sodium intake would remain above the CDRR thresholds.

## Figures and Tables

**Figure 1 nutrients-13-00779-f001:**
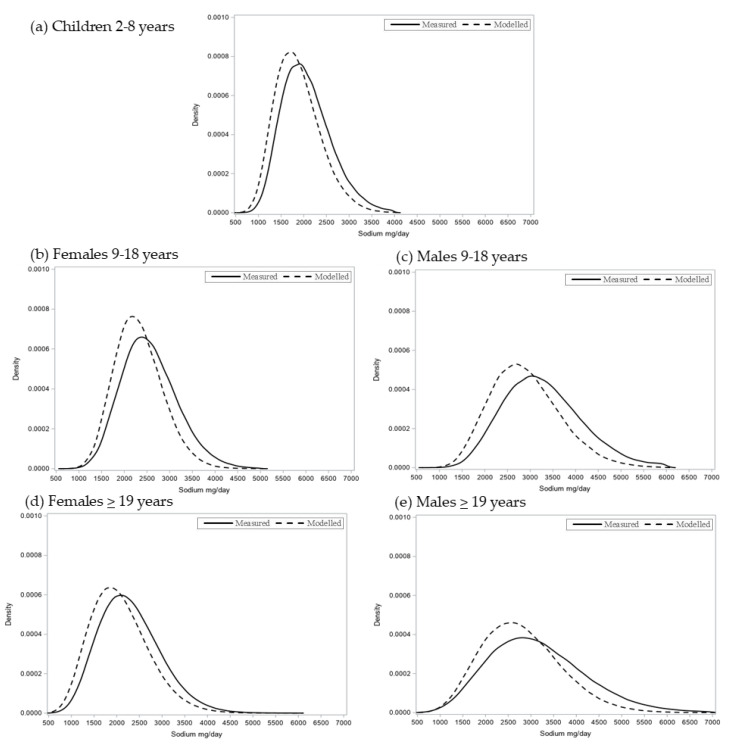
Estimated distribution of usual sodium intake (mg/day) reported at measured (solid line) and modelled full compliance to Health Canada’s voluntary Sodium Reduction Guidance (SRG) targets for processed food (dotted line) in (**a**) children 2 to 8 years, (**b**) females 9–18 years, (**c**) females ≥19 years, (**d**) males 9–18 years and (**e**) males ≥19 years from the 2015 Canadian Community Health Survey—Nutrition (*n* = 19,645). Results are generated using the NCI method and applying 100 Monte Carlo simulations. Results are weighted and adjusted for age, sex (children 2–8 years), misreporting, sequence of recall and weekend/weekday recall. Distributions are trimmed according to Statistics Canada vetting procedures.

**Table 1 nutrients-13-00779-t001:** Unadjusted weighted mean sodium intake (mg/day) across main study variables in Canadians aged 2 to 18 years and 19 years and older in the 2015 Canadian Community Health Survey-Nutrition (*n* = 19,645).

	2 to 18 Years		≥19 Years	
		Sodium (mg/day)		Sodium (mg/day)
	*n*	Mean (95% CI) (95% CI)	*n*	Mean (95% CI) (95% CI)
Sex				
Males	3072	2820 (2730, 2910)	6414	3179 (3088, 3269)
Females	3054	2324 (2253, 2394)	7105	2295 (2238, 2352)
Dietary Reference Intake Groups				
Male and Female				
2–3 years	948	1750 (1634, 1865)		
4–8 years	1223	2287 (2197, 2378)		
Males				
9–13 years	1044	2937 (2807, 3067)		
14–18 years	934	3520 (3327, 3713)		
19–30 years			881	3560 (3300, 3821)
31–50 years			2070	3264 (3098, 3430)
51–70 years			2227	3046 (2919, 3174)
≥71 years			1236	2649 (2533, 2765)
Females				
9–13 years	966	2582 (2468, 2695)		
14–18 years	1011	2498 (2353, 2644)		
19–30 years			892	2387 (2185, 2588)
31–50 years			2273	2419 (2315, 2523)
51–70 years			2405	2194 (2121, 2267)
≥71 years			1535	2100 (2013, 2186)
Household Education				
Less than high school	190	2738 (2391, 3085)	1556	2670 (2502, 2838)
High school only	939	2575 (2443, 2707)	2722	2720 (2597, 2843)
Trade, college, etc.	2458	2670 (2571, 2769)	4932	2810 (2710, 2911)
Bachelor degree, etc.	2539	2477 (2390, 2563)	4309	2685 (2596, 2775)
Income				
Quintile 1	1190	2499 (2379, 2618)	2829	2566 (2406, 2726)
Quintile 2	1158	2509 (2397, 2621)	2870	2632 (2528, 2736)
Quintile 3	1407	2644 (2510, 2778)	2866	2800 (2692, 2908)
Quintile 4	1271	2496 (2366, 2625)	2381	2758 (2636, 2879)
Quintile 5	1100	2764 (2601, 2926)	2573	2916 (2784, 3049)
Food security				
Food insecure	772	2753 (2531, 2976)	1270	2679 (2484, 2873)
Food secure	5354	2551 (2491, 2611)	12,249	2742 (2683, 2801)
Misreporting				
Under reporter	1144	1828 (1724, 1933)	4374	1842 (1786, 1898)
Plausible reporter	2530	2724 (2653, 2796)	6551	3050 (2980, 3120)
Over reporter	671	3658 (3478, 3838)	846	4898 (4649, 5146)
Unclassified	1781	2367 (2254, 2480)	1748	2727 (2565, 2889)
Energy intake				
Males (kcal/day)	3072	1995 (1943, 2046)	6414	2173 (2123, 2223)
Females (kcal/day)	3054	1674 (1626, 1721)	7105	1590 (1559, 1621)

**Table 2 nutrients-13-00779-t002:** Adjusted mean sodium intake and mean differences by socioeconomic position as measured and modelled full compliance to Health Canada’s voluntary Sodium Reduction Guidance (SRG) targets in children 2 to 8 years old in the 2015 Canadian Community Health Survey—Nutrition (*n* = 2171).

	Measured Sodium Intake (mg/day)		Modelled Sodium Intake (mg/day)	
	Mean (95% CI)	Difference (95% CI)	Mean (95% CI)	Difference (95% CI)
Household Education				
Less than high school	2379 (1822, 2936)	358 (−201, 916)	1959 (1685, 2233)	72 (−204, 347)
High school only	2233 (2034, 2432)	212 (5, 419)	1949 (1843, 2055)	61 (−48, 171)
Trade, college, etc.	2126 (2045, 2208)	105 (9, 201)	1904 (1818, 1990)	16 (−74, 106)
Bachelor degree, etc.	2021 (1948, 2095)	Ref.	1888 (1831, 1944)	Ref.
Income				
Quintile 1	2151 (2026, 2276)	170 (−1, 342)	1914 (1826, 2002)	49 (−77, 175)
Quintile 2	2070 (1949, 2192)	90 (−68, 249)	1952 (1844, 2061)	87 (−49, 224)
Quintile 3	2132 (2015, 2249)	151 (3, 299)	1917 (1822, 2011)	52 (−71, 174)
Quintile 4	2060 (1981, 2140)	80 (−54, 214)	1842 (1774, 1911)	−23 (−141, 96)
Quintile 5	1980 (1863, 2098)	Ref.	1865 (1767, 1963)	Ref.
Food Security				
Food insecure	2335 (2140, 2531)	271 (75, 468)	1976 (1851, 2102)	83 (−45, 212)
Food secure	2064 (2004, 2124)	Ref.	1893 (1837, 1949)	Ref.

Results are adjusted for continuous age, sex, total daily energy intake (kcal/day), and misreporting.

**Table 3 nutrients-13-00779-t003:** Sex-specific adjusted mean sodium intake and mean difference by socioeconomic position as measured and modelled full compliance to Health Canada’s voluntary Sodium Reduction Guidance (SRG) targets in Canadians 9–18 years (*n* = 3955) and >19 years (*n* = 13,519) in the 2015 Canadian Community Health Survey—Nutrition.

	Males				Females			
	Measured Sodium Intake (mg/day)		Modelled Sodium Intake (mg/day)		Measured Sodium Intake (mg/day)		Modelled Sodium Intake (mg/day)	
	Mean (95% CI)	Difference (95% CI)	Mean (95% CI)	Difference (95% CI)	Mean (95% CI)	Difference (95% CI)	Mean (95% CI)	Difference 95% CI)
9–18 Years								
Household Education								
Less than high school	3249 (2949, 3550)	148 (−161, 456)	2562 (2272, 2853)	−211 (−509, 87)	2740 (2498, 2981)	246 (−1, 493)	2442 (2202, 2682)	96 (−149, 341)
High school only	3115 (2939, 3290)	13 (−175, 201)	2785 (2643, 2926)	12 (−129, 152)	2523 (2394, 2652)	29 (−127, 185)	2256 (2146, 2365)	−90 (−227, 46)
Trade, college, etc.	3223 (3084, 3362)	121 (−34, 277)	2780 (2655, 2905)	7 (118, 131)	2535 (2430, 2641)	41 (−86, 169)	2250 (2156, 2344)	−96 (−216, 24)
Bachelor degree, etc.	3102 (2974, 3229)	Ref.	2773 (2651, 2896)	Ref.	2494 (2387, 2601)	Ref.	2346 (2243, 2449)	Ref.
Income								
Quintile 1	3052 (2892, 3211)	−224 (−442, −5)	2718 (2573, 2863)	−56 (−244, 133)	2521 (2378, 2663)	15 (−172, 201)	2366 (2238, 2495)	104 (−63, 271)
Quintile 2	3069 (2923, 3214)	−207 (−423, 10)	2814 (2666, 2962)	41 (−159, 240)	2617 (2464, 2771)	111 (−90, 312)	2375 (2232, 2519)	113 (−79, 304)
Quintile 3	3262 (3105, 3419)	−14 (−245, 217)	2803 (2679, 2928)	30 (153, 213)	2532 (2403, 2660)	26 (−143, 194)	2267 (2153, 2382)	5 (−149, 159)
Quintile 4	3118 (2947, 3290)	−157 (−391, 78)	2731 (2550, 2912)	−43 (−245 160)	2421 (2312, 2530)	−85 (−249, 79)	2185 (2078, 2292)	−78 (−238, 83)
Quintile 5	3275 (3078, 3472)	Ref.	2773 (2600, 2947)	Ref.	2506 (2370, 2643)	Ref.	2262 (2129, 2396)	Ref.
Food Security								
Food insecure	3138 (2937, 3339)	−19 (−227, 189)	2711 (2547, 2875)	−66 (−232, 100)	2636 (2469, 2803)	122 (−52, 296)	2295 (2153, 2436)	−4 (−149, 140)
Food secure	3157 (3052, 3263)	Ref.	2777 (2670, 2883)	Ref.	2514 (2434, 2593)	Ref.	2299 (2224, 2374)	Ref.
≥19 Years								
Household Education								
Less than high school	3400 (3186, 3614)	399 (172, 626)	2918 (2748, 3087)	148 (−30, 327)	2412 (2306, 2517)	122 (6, 239)	2067 (1981, 2152)	−45 (−141, 51)
High school only	3177 (3045, 3309)	176 (26, 325)	2751 (2641, 2861)	−18 (−143, 107)	2358 (2266, 2451)	69 (−42,180)	2110 (2032, 2188)	−2 (−97, 94)
Trade, college, etc.	3076 (2969, 3184)	75 (−51, 201)	2745 (2654, 2837)	−24 (−130, 82)	2353 (2273, 2433)	64 (−38, 166)	2119 (2047, 2191)	7 (−81, 94)
Bachelor degree, etc.	3001 (2876, 3126)	Ref.	2769 (2660, 2878)	Ref.	2289 (2199, 2380)	Ref.	2112 (2035, 2188)	Ref.
Income								
Quintile 1	3113 (2933, 3293)	82 (−126, 290)	2811 (2667, 2954)	126 (−40, 292)	2305 (2206, 2404)	−41 (−184, 103)	2105 (2021, 2189)	21 (−111, 154)
Quintile 2	3083 (2949, 3217)	52 (−118, 222)	2781 (2668, 2893)	96 (−49, 241)	2313 (2225, 2401)	−33 (−171, 106)	2110 (2034, 2185)	26 (−95, 146)
Quintile 3	3129 (2999, 3259)	98 (−56, 252)	2814 (2708, 2919)	129 (−2, 260)	2411 (2312, 2509)	65 (−82, 211)	2153 (2072, 2234)	70 (−57, 195)
Quintile 4	3120 (2984, 3256)	89 (−74, 251)	2770 (2651, 2889)	86 (−56, 227)	2321 (2220, 2423)	−25 (−182, 133)	2092 (2007, 2177)	8 (−131, 147)
Quintile 5	3031 (2887, 3175)	Ref.	2685 (2555, 2814)	Ref.	2346 (2222, 2470)	Ref.	2084 (1969, 2199)	Ref.
Food Security								
Food insecure	3137 (2910, 3365)	48 (−186, 282)	2596 (2423, 2770)	−188 (−365, −11)	2276 (2168, 2385)	−67 (−172, 38)	2007 (1912, 2101)	−112 (−206, −18)
Food secure	3089 (2996, 3182)	Ref.	2784 (2703, 2865)	Ref.	2343 (2280, 2406)	Ref.	2119 (2064, 2175)	Ref.

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
