# Peer review of "The Equity and Effectiveness of Achieving Canada’s Voluntary Sodium Reduction Guidance Targets: A Modelling Study Using the 2015 Canadian Community Health Survey—Nutrition"

_nutrients, 2021, doi:10.3390/nu13030779_

Round 1

Reviewer 1 Report

The Introduction section is rather superficial. Authors should perform a deeper analysis of the literature that will serve as a basis to justify the present study.

Tables must be formatted according to the journal's rules. For example, Table 3 has text parts cut off.

Figure 1 should be improved. It becomes complicated to read the data.

Reviewer 2 Report

Authors used results from 2015 Canadian Community Health Survey and compared average sodium intake level across socioeconomic position (SEP) groups. And also calculated their estimated sodium intake when the sodium contents of the foods are reduced according to Sodium Reduction Guidance (SRG), and examined whether the sodium reduction intended in SRG would reduce the inequity among SEPs.

INTRODUCTION
Higher sodium intake level among lower socioeconomic groups has been reported in many countries. In Canada, what kind of characteristics accounts for the higher sodium intake in lower SEP, ie., higher body weight (more food intake), more high salt foods, less no-salt foods?

METHODS
L156 "Adjusted household income quintiles were estimated"
Please address how the adjusted household income was calculated. Was it adjusted with number of household members, local area, or other factors? Dose the formula indicate some explanation to the inconsistent association of household income quintiles and sodium intake?

L193-194 "Reported foods were regrouped into 15 categories and 94 subcategories outlined in the SRG targets."
Only processed/salt added foods are listed in the SRG and there should have been foods reported in the 24 hour recalls that were not categorized into the 15 SRG food groups. Were the amount and sodium intake from those foods (not in SRG list) calculated?

RESULTS
L290 "aduls in households with 'less than high school' education had lower mean,,,," , should be "higher" instead of "lower".

Supplementary Tables S2, 3, 4
Many of "modeled intake" values for phase I are higher than measured intake. Why does it happen? Were the phase I Na contents higher than Na value in the original food database for some foods?

DISCUSSION
Authors showed that inequity in sodium intake among SEPs will be reduced with sodium reduction in processed foods according to SRG. Would it be possible to present some food categories that attribute the inequity reduction? Acceleration of sodium reduction in such foods would be beneficial for inequity reduction.

Round 2

Reviewer 1 Report

The authors have addressed all my comments. So, the manuscript can now be published.